# Digestive Amyloidosis Trends: Clinical, Pathological, and Imaging Characteristics

**DOI:** 10.3390/biomedicines12112630

**Published:** 2024-11-17

**Authors:** Sandica Bucurica, Andreea-Simona Nancoff, Miruna Valeria Moraru, Ana Bucurica, Calin Socol, Daniel-Vasile Balaban, Mihaela Raluca Mititelu, Ionela Maniu, Florentina Ionita-Radu, Mariana Jinga

**Affiliations:** 1Department of Internal Medicine and Gastroenterology, Carol Davila University of Medicine and Pharmacology, 020021 Bucharest, Romania; sandica.bucurica@umfcd.ro (S.B.); vasile.balaban@umfcd.ro (D.-V.B.); mariana.jinga@umfcd.ro (M.J.); 2Department of Gastroenterology, University Emergency Central Military Hospital “Dr. Carol Davila”, 024185 Bucharest, Romania; andreea-simona.nancoff@rez.umfcd.ro (A.-S.N.); miruna-valeria.moraru@drd.umfcd.ro (M.V.M.); 3General Medicine, Carol Davila University of Medicine and Pharmacology, 020021 Bucharest, Romania; ana.bucurica@stud.umfcd.ro (A.B.); calin.socol@stud.umfcd.ro (C.S.); 4Department of Nuclear Medicine, University of Medicine and Pharmacy Carol Davila, 020021 Bucharest, Romania; 5Department of Nuclear Medicine, University Emergency Central Military Hospital, 010825 Bucharest, Romania; 6Department of Mathematics and Informatics, Faculty of Sciences, Lucian Blaga University Sibiu, 550012 Sibiu, Romania; ionela.maniu@ulbsibiu.ro; 7Research Team, Pediatric Clinical Hospital Sibiu, 550166 Sibiu, Romania

**Keywords:** digestive amyloidosis, amyloid deposits, digestive motility, amyloid light chains, gastrointestinal amyloidosis

## Abstract

Amyloidosis is a group of diseases characterized by the extracellular deposition of abnormally folded, insoluble proteins that lead to organ dysfunction. While it commonly affects the cardiovascular system, gastrointestinal (GI) tract involvement is undetermined. Recent research has focused on understanding the pathophysiology, diagnostic challenges, and therapeutic approaches to GI amyloidosis, particularly in systemic amyloid light-chain (AL) and amyloid A (AA) forms. GI manifestations can include motility disorders, bleeding, and, in severe cases, bowel obstruction. This review highlights the importance of the early recognition of digestive symptoms and associated imagistic findings in GI amyloidosis by analyzing the research that included clinical, pathological, and endoscopic approaches to amyloidosis. A systematic search of the PubMed and Scopus databases identified 19 relevant studies. Our findings showed that amyloid deposits commonly affect the entire GI tract, with AL amyloidosis being the most predominant form. Endoscopic evaluations and biopsy remain key diagnostic tools, with Congo Red staining and mass spectrometry being used to confirm amyloid type. Although progress has been made in diagnosis, the absence of targeted therapies and the indistinct nature of GI symptoms continue to be challenging.

## 1. Introduction

Amyloidosis refers to a group of diseases caused by the abnormal production of proteins, leading to the accumulation of insoluble protein deposits in various tissues [1]. These deposits disrupt normal organ function and are identifiable by their ability to bind to Congo Red stain, which produces a distinctive greenish color when viewed under polarized light microscopy [1,2]. Clinically, amyloidosis manifests as a miscellaneous illness that affects a wide variety of organs, the cardiovascular system being the most frequently involved (75%) [3]. The kidneys, soft tissues, and peripheral or autonomic nervous system are other possible sites for fibrillary protein accumulation [4] The gastrointestinal tract (GI) is an uncommon site, and the liver is the most affected organ (15%) [3,4]. Over the years, there has been constant interest in amyloidosis and its involvement in the GI tract. Hence, much research has been committed to better understanding its pathophysiology, diagnostic challenges, and therapeutic approaches [5]. The GI involvement of amyloidosis has been described in amyloid light-chain (AL) systemic form, as well as both familial types of amyloidosis and secondary reactive forms (the accumulation of the serum amyloid A protein—AA amyloidosis) [6,7,8]. There are a wide range of studies focusing on how amyloid deposits can lead to complications such as motility disorders, GI bleeding, malabsorption, and, in some cases, even bowel obstruction [9,10,11,12]. A series of publications by Rubinow A. et al. and Burakoff R. et al. revealed, at that time, that more than half of patients diagnosed with one form of amyloidosis had esophageal motility abnormalities [13,14]. Gonzalez J et al. also presented dysphagia arising from motility changes as an uncommon complication of AL amyloidosis [15]. Moreover, these abnormalities concerning GI motility were also recorded in small bowel movements, as proved by Wixner J et al. [16]. The abnormal buildup of proteins leads to malabsorption or, in severe cases, even bowel obstruction [17]. The mechanism is still not fully comprehended. Still, it is believed that it may arise from bacterial overgrowth secondary to dysmotility, the accumulation of amyloid in smooth muscles, or changes in the neuroendocrine cells that line the GI tract [18,19,20]. Even though the last decades have uncovered substantial progress in the diagnosis and treatment of amyloidosis, especially AL amyloidosis, the lack of target therapy for GI involvement and the equivocal clinical manifestations continue to make this disease a challenge in the field of gastroenterology [21]. In their study, Yen et al. tried to define further the role of gastroenterologists in managing GI amyloidosis symptoms. They emphasized the importance of recognizing specific symptoms as part of amyloidosis rather than classifying them as visceral hypersensitivity [22]. Therefore, our review aims to highlight the importance of the early recognition of digestive symptoms, signs, and imagistic findings associated with GI amyloidosis.

## 2. Materials and Methods

We systematically searched two databases (PubMed and Scopus) from the beginning to 15 June 2024. Our search aimed to find as much information as possible about the GI involvement of amyloidosis (including liver and pancreas afflictions), changes associated with amyloidosis, clinical and imagistic findings, and diagnostic methods. The following search key was used: (amyloid[MeSH Terms]) OR amyloidosis OR “amyloid fibrils” OR (amyloidosis[MeSH Terms])) AND ((digestive system[MeSH Terms]) OR (liver[MeSH Terms]) OR (pancreas[MeSH Terms]) OR intestinal). The articles yielded by the advanced search were downloaded into Rayyan (Cambridge, MA, USA). Duplicates were automatically removed and then manually removed afterward. The articles that met the inclusion criteria presented clinical or observational data on amyloid deposition in the GI tract, the type of amyloidosis, clinical features, and diagnostic methods (Figure 1). Studies were screened for detailed information on patient demographics, associated symptoms, endoscopic lesion characteristics, and diagnostic procedures. Data from each study were extracted based on 30 predefined factors, consisting of the title of each publication, the first author, the year of publication, the journal of publication, the type of article, the number of patients, the sex of the participants, the ages of the participants, the evolution of symptoms (months), digestive symptoms and signs, other symptoms and signs, relevant biologic markers, paraclinical diagnosis, endoscopic lesions (size, type, site), another imagistic and functional test (lesion size, site, type), the tissue acquisition site and method, histological aspects, the type of disease (localized or systemic), other organs involved and which ones, and the type of amyloidosis. Based on these, we summarized the information that we gathered in a structured table for analysis: Table 1. An independent researcher (S.B.) conducted the selection, and two others (A.B. and C.S.) solved disagreements. Two other independent evaluators (A-S.N. and M.V.M) extracted data. The data included the first author, publication year, total number of trial participants, type of amyloidosis, diagnostic methods, clinical features, and imagistic and functional test findings. We considered eligible articles that met the following inclusion criteria: human subject studies; publications written in English with the full text available; observational, cohort, randomized controlled, and cross-sectional studies focusing on amyloidosis and its digestive implications; and studies focusing on clinical and imagistic features. We excluded non-human studies, publications in languages other than English, studies with the full text unavailable, and case reports, case series, and narrative reviews. Studies focusing on amyloidosis involving another system than the GI tract were also excluded, as well as those on the endocrine function of the pancreas.

This systematic review was registered in PROSPERO 2024 CRD42024546611. The review is available at https://www.crd.york.ac.uk/prospero/display_record.php?ID=CRD42024546611, accessed on 24 October 2024, where the study protocol can be accessed.

Visual network mapping (performed by I.M.) was used to provide an overview of characteristics, research trends, and emerging areas in the amyloidosis research field by utilizing VOSviewer software version 1.6.16 (VanEck and Waltman, Center for Science and Technology Studies of Leiden University, Leiden, The Netherlands [23]), a widely used bibliometric tool in medicine [24,25,26,27,28]. This analysis was also utilized in other research areas and applied to co-occurrence, co-authorship, co-citation references, and visualization networks [29,30,31,32,33,34].

Our aim was to evaluate whether suspicion of amyloidosis can be raised when patients present with certain symptoms and whether there is a connection between the clinical findings and specific endoscopy and functional test findings.

## 3. Results

Nineteen publications were analyzed based on the search strategy. These studies examined various amyloidosis-related organ involvements, diagnostic techniques, and disease outcomes. A significant portion of the publications included retrospective observational studies that focused on amyloid deposits in different systems, such as the GI tract, heart, kidneys, and even the nervous system. Several studies, such as those by Hagen CE et al. and Kobayashi H. et al., involved large cohorts of patients in whom they investigated systemic amyloid involvement in different segments of the GI tract [8,35]. A few retrospective studies, such as that of Krauß L. et al., assessed the progression and incidence of amyloidosis over time [36]. Most studies focused on systemic amyloidosis and common diagnostic methods, such as biopsy, Congo Red (CR) staining, and immunohistochemistry. Table 1 summarizes data on study design, organ involvement, biopsy techniques, amyloid typing methods, and clinical, imagistic, and functional test outcomes.

Based on our research, we observed that almost all studies had GI tract involvement, with AL amyloidosis (primary) being the most predominant form affecting the GI system, followed by AA amyloidosis (secondary) (Table 1).

**Table 1 biomedicines-12-02630-t001:** Main clinical, imagistic, and functional test findings in different types of GI amyloidosis.

Organ	Author and Year	Study Population	Type of Amyloidosis	Pathological Diagnosis	Clinical Features	Imagistic and Functional Test Findings *
**Esophagus** **Stomach** **Small intestine** **Colon**	Hagen CE et al. (2023) [8]	2511 patients	Primary, secondary and, familial amyloidosis	Biopsies taken from the upper and lower GI tract, Congo Red stained, showed amyloid deposits. Mass spectrometry-based proteomics for amyloid typing.	Diarrhea GI bleed Abdominal pain Weight loss Reflux Abnormal imaging studies Nausea Vomiting Chest pain Dysphagia	Endoscopy—inflammation/ulceration/friability/hyperemia, amyloidoma, mucosal atrophy, abnormal esophageal motility, esophageal varices, and thickened rectal fold. No lesions or changes in the mucosa were also reported.
**Stomach** **Small intestine** **Colon** **Rectum** **Liver**	Krauß L et al. (2023) [36]	63 patients	Primary, secondary and, familial amyloidosis	Biopsies taken from the upper and lower GI tract, Congo Red stained, showed amyloid deposits.	Diarrhea Abdominal pain	Endoscopy—erosions, ulcers, inflammation, polypoid structures, polyps, hematomas, vascular malformations, and diverticulum.
**Colon** **Rectum**	Nakov R et al. (2020) [37]	21 patients	Familial amyloid polyneuropathy	Biopsies taken from the lower GI tract, Congo Red stained, showed amyloid deposits. Fecal calprotectin levels assessment	Abdominal pain Nausea, Vomiting Weight loss Constipation Diarrhea Alternating diarrhea/constipation	Endoscopy findings—N/A ** Abdominal ultrasonography—small bowel dilatation, stasis, and enhanced peristalsis.
**Stomach** **Duodenum** **Rectum**	Matsuda M et al. (2014) [9]	202 patients	AL amyloidosis myeloma-associated amyloidosis	Immunohistochemical analyses of the deposited amyloid fibril proteins using antibodies against four common different amyloid fibrils proteins Serum and urine immunofixation	GI bleeding—rectal or melena	Endoscopy—solitary tumors (amyloidomas) in the stomach, multiple nodular lesions and erosions, mucosal friability, and easily bleeding. CT—marked thickening of the wall in the stomach, especially in the distal portion.
**Rectum**	Bektaş M et al. (2009) [38]	31 patients	Familial amyloid polyneuropathy	Biopsies taken from the lower GI tract, Congo Red stained, showed amyloid deposits	Dyspepsia—heartburn, dysphagia, noncardiac chest pain, bloating, and epigastric pain	Endoscopy—esophagitis Los Angeles A and B, sliding-type hiatal hernia, Barrett’s esophagus. Esophageal motility studies—abnormalities regarding esophageal motility.
**Small intestine**	Kala Z et al. (2007) [39]	7 patients	Primary and secondary amyloidosis.	Biopsies taken from the upper and lower GI tract, Congo Red stained, showed amyloid deposits.	Severe dyspepsia	Endoscopy—N/A **. Enteroclysis—polypoid-like structures and thumb-printing sign. CT—tumor-like lesions, diffuse accumulation of amyloid in the liver. Abdominal ultrasonography—thickening of the intestine’s wall.
**Stomach** **Duodenum** **Colon** **Liver**	James DG et al. (2007) [20]	19 patients	Primary amyloidosis	Biopsies taken from the upper and lower GI tract, Congo Red stained, showed amyloid deposits.	Gastrointestinal bleeding Abdominal pain Altered bowels Abnormal gut motility Weight loss	Endoscopy—polypoid-like submucosal hematoma, ulceration/erosions, inflammation, and edema.
**Small intestine**	Hayman et al. (2001) [18]	19 patients	Primary amyloidosis	Biopsies taken from the upper GI tract, Congo Red stained, showed amyloid deposits. Bone marrow biopsy. Presence of elevated serum or urine M-protein	Diarrhea Anorexia Steatorrhea Vomiting Fecal incontinence Dysphagia Dysgeusia weight loss Hepatomegaly	Endoscopy—esophagitis, gastro-duodenitis, ulcer, pseudo-obstruction, gastroparesis, polyps’ inflammation, and colitis. Normal aspects of the mucosa from either the upper or lower GI were also found.
**Esophagus** **Stomach** **Small intestine** **Colon** **Rectum**	Bilezikçi B et al. (2000) [40]	78 patients	Primary and secondary amyloidosis	Biopsies taken from the upper and lower GI tract, Congo Red stained, showed amyloid deposits. Serum amyloid A (SAA) levels.	Vomiting Diarrhea Abdominal pain Nausea Dyspepsia Chronic gastritis	Endoscopy—chronic gastritis, congestion, edema, and inflammation. In the rectum, there were mild crypt distortions, inflammation, and severe ulcerations present.
**Stomach** **Duodenum** **Colon** **Rectum**	Yoshimatsu S et al. (1998) [41]	9 patients	Familial amyloid polyneuropathy	Biopsies taken from the upper and lower GI tract, Congo Red stained, showed amyloid deposits. Radioimmunoassay and polymerase chain reaction	N/A	Endoscopy—fine granular appearance, lack of luster, and mucosal friability.
**Stomach** **Duodenum**	Kobayashi H et al. (1996) [35]	407 patients	Secondary amyloidosis	Biopsies taken from the upper GI tract, Congo Red stained, showed amyloid deposits.	Diarrhea Nausea Abdominal pain Anorexia	Endoscopy—gastritis, duodenitis, polypoid lesions, ulcers, erosions, and mucosal friability.
**Esophagus** **Duodenum** **Small intestine** **Colon** **Rectum**	Tada et al. (1994) [42]	49 patients	Primary, secondary, and AH-type amyloidosis	Biopsies taken from the upper and lower GI tract, Congo Red stained, showed amyloid deposits. Antibody staining for amyloidosis typing	Abdominal pain Abdominal distension Constipation Gi bleeding Dysphagia Anorexia Nausea Vomiting	Endoscopy—granular elevations, polypoid protrusions, thickening of the folds Radiologic investigations—coarse mucosa pattern thickening of the valvulae coniventes, marked small and large intestine dilatation, and motility abnormalities.
**Jejunum**	Tada S et al. (1994) [43]	30 patients	Primary, and secondary amyloidosis	Immunohistochemical analysis using anti-AA, anti-pre albumin, and anti-β2 micro-globulin antibodies. Biopsies taken from the upper and lower GI tract, Congo Red stained, showed amyloid deposits.	Diarrhea Malabsorption Intestinal obstruction Melena Occult GI bleeding	Endoscopy—fine granular appearance, erosions, and mucosal friability, thickening of the valvulae conniventes, multiple polypoid structures, shallow ulcers, and amyloidoma.
**Small intestine**	Tada S et al. (1991) [44]	26 patients	Primary, secondary, and familial amyloidosis	Biopsies taken from the jejunum showed amyloid A protein deposits	Abdominal pain, Diarrhea Obstruction Malnutrition	Endoscopy- fine granular appearance, multiple erosions, polypoid structures, and nodular lesions with sizes varying from 1 to 10 mm.
**Esophagus** **Stomach** **Duodenum** **Colon** **Rectum**	Tada S et al. (1990) [45]	37 patients	Primary and secondary amyloidosis	Biopsies taken from the upper and lower GI tract, Congo Red stained, showed amyloid deposits.	Abdominal pain, Hematemesis, Diarrhea, Vomiting	Endoscopy—fine granular appearance, polypoid structures, erosions, ulcerations, and mucosal friability.
**Jejunum**	Feurle GE et al. (1987) [46]	7 patients	Familial amyloid polyneuropathy	Biopsies taken from the upper GI tract, Congo Red stained, showed amyloid deposits.	Diarrhea Steatorrhea	Endoscopy—normal aspect of the mucosa, no signs of inflammation.
**Stomach** **Rectal**	Yamada M et al. (1985) [47]	21 patients	Primary and secondary amyloidosis	Biopsies taken from the upper and lower GI tract, Congo Red stained, showed amyloid deposits.	Anorexia Nausea Vomiting Constipation Diarrhea GI bleeding Malabsorption Pseudo-obstruction	Specimen resection—mucosal atrophy, erosions, ulcers.
**Rectum**	Burakoff R et al. (1985) [14]	13 patients	Familial amyloid polyneuropathy	Biopsies taken from the lower GI tract, Congo Red stained, showed amyloid deposits.	Fecal incontinence, Diarrhea Steatorrhea	Endoscopy findings—N/A **. Esophageal motility studies—abnormalities regarding esophageal motility.
**Stomach** **Small intestine up to Treitz ligament** **Rectum**	Steen L et al. (1983) [48]	21 patients	Familial amyloid polyneuropathy	Biopsies taken from the upper and lower GI tract, Congo Red stained, showed amyloid deposits.	Constipation Bout of diarrhea	Endoscopy—normal aspect of the mucosa, no signs of inflammation.

* Endoscopy/computed tomography/ultrasonography/enteroclysis/manometry; ** N/A—not available; CT—computed tomography; AH—immunoglobulin heavy-chain amyloidosis; AA—amyloid A.

AL amyloidosis was often seen in patients with systemic involvement, with the AL lambda chain being the most common in every region of the GI tract (stomach, duodenum, jejunum/ileum, large intestine, and rectum). AA amyloidosis was common in patients with chronic inflammatory diseases like rheumatoid arthritis [40,49,50]. In some cases, AA was secondary to conditions like tuberculosis or chronic infections [43]. Even in those studies published almost forty years ago, the main symptoms that prompted imagistic or functional studies were ambiguous and non-specific. Changes in bowel movement (diarrhea or constipation), vague abdominal pain, dyspepsia, nausea, and sometimes even bleeding (occult GI bleeding, hematemesis, or melena) were the main clinical manifestations reported [9,40,44]. In these cases, endoscopic evaluations revealed distinctive patterns of amyloid deposition in various GI regions. Granular densities, erosions, and polypoid protrusions, ranging in size from 1 to 10 mm, were observed in the jejunum [44]. Other findings seen in patients with advanced stages of systemic amyloidosis included nodular lesions, submucosal hematomas, and ulcerations, thus suggesting a systemic progression [20]. In FAP patients, granular lesions and a lack of mucosa luster, particularly in the stomach and duodenum, were reported [41]. Apart from the most common findings (granular deposits, ulcerations, etc.) reported in most of the studies that we analyzed, Hagen CE et al., Matsuda M et al., and Tada S et al. also noted the presence of amyloidomas. Essentially, these structures are rare tumor-like deposits of amyloidogenic proteins occurring in various body regions. They were described in brain tissue, pharynx, soft tissues, pulmonary tissue, and ganglia [51,52,53,54,55]. Several case reports describe the complications associated with these abnormal buildups of amyloid deposits in the GI tract, especially GI bleeding [56]. Mass spectrometry-based proteomic typing via endoscopic biopsy effectively identified multiple amyloid types across the entire GI tract [8]. Endoscopy and manometry were used in patients with Transthyretin Amyloidosis (ATTR) and Familial Mediterranean Fever (FMF) to reveal colonic mucosa abnormalities and lower esophageal sphincter hypomotility [37,38]. This emphasizes the role of imagistic and functional tests for amyloid detection, particularly in the early stages of GI involvement.

Histopathological findings from various studies confirmed the presence of amyloid deposits in different regions of the GI tract, with most studies utilizing biopsy methods to confirm amyloid deposition. The biopsy with Congo Red (CR) staining remains the most reliable method for identifying amyloidogenic proteins across various tissue types [57]. Typically, a standard hematoxylin and eosin (HE) stain reveals extracellular, eosinophilic, homogeneous, and amorphous structures indicative of amyloid deposition [58]. A CR stain is subsequently applied to confirm the diagnosis, which displays green birefringence characteristic of amyloid structures when examined under polarized light microscopy [57]. Further methods, such as skin biopsy, either performed by punch or excision, or abdominal fat biopsy, also performed by punch or excision, will be discussed later. In the small intestine, amyloid deposits were present both in the mucosa and the submucosa, with additional deposits in the walls of submucosal vessels [45]. The liver, pancreas, and kidneys also showed significant amyloid deposition in cases of systemic amyloidosis. Amyloid deposits were identified in the cortical interstitium of the kidneys and the periportal and pericentral vein regions in the liver [59]. Both vascular and interstitial spaces showed deposits with various degrees of thickness in submucosal vessels in cases with systemic involvement [40]. In some cases of ATTR amyloidosis, amyloid deposits in the rectal and colonic mucosa were associated with neutrophilic granulocytic infiltration [59]. Tissue biopsy and advanced diagnostic tools, such as mass spectrometry and immunohistochemistry, were used to identify amyloid deposits and their specific protein constituents. MALDI-Mass spectrometry imaging (MALDI-MSI), a technique that combines matrix-assisted laser desorption/ionization with mass spectrometry to visualize the spatial distribution of molecules in tissue samples, was used to identify vitronectin as a common component in amyloid deposits across multiple organs, including the liver, kidney, heart, and brain [60]. Amyloid deposits in the lung and gastrointestinal tract were explored using MALDI-MSI, correlating peptide signatures with ApoE and ApoA1 in AL amyloidosis cases [61]. Also, using immunohistochemistry and proteomics, amyloid typing across multiple organ sites was compared. This validates the reproducibility of amyloid typing across the myocardium, kidney, liver, and small intestine, confirming that proteomics is reliable in systemic amyloidosis involving multiple organs [62]. Moreover, to avoid repeated biopsies, Ezawa N et al. showed that 11C-PiB PET imaging could be a non-invasive option for visualizing amyloid deposits as the accumulation of amyloid in the stomach seen on the images correlated with biopsy-confirmed amyloid deposition. However, further studies are needed to confirm its clinical utility [63].

Even though amyloidosis is regarded as a heterogeneous disease, the amount of research carried out in the last few decades seems to shed light on its complexity. Endoscopic and imagistic techniques have helped to diagnose this disease, as our literature search proves [11,63,64]. Moreover, whenever amyloidosis is confirmed, endoscopic surveillance should be performed to assess the progression and possible disease-associated complications. Several studies have also reported abdominal or skin fat biopsy as an alternative to repeated endoscopic procedures [65,66].

Abdominal fat pad excision biopsy (FPEB) was reported to have good sensitivity for AL amyloidosis and higher accuracy, as the tissue sample was larger [67]. However, Muchtar E et al. proved that overusing organ biopsies, especially in AL amyloidosis, may fail to recognize the early stages of this disease, but further studies are needed to establish their results [66]. Others like Rokke HP et al. also reported the good precision of abdominal fat biopsy in patients with suspected ATTR amyloidosis [68]. Like these methods, abdominal skin biopsy, either by excision or puncture, was also evaluated as a potential minimally invasive technique for diagnosing amyloidosis [69]. Wu B et al. and Pinton S et al. also reported optimal sensitivity and showed promising results regarding amyloidosis diagnosis, especially for the AL and ATTR types [69,70]. Although large cohort studies need to be established further, these results show encouraging prospects for future research.

We also performed a bibliometric review of all documents retrieved from the Scopus (2079 documents) and PubMed (497) databases to provide an overview of characteristics, research trends, and emerging areas in the researched field. Results from the VOSviewer maps of keywords and terms from title and abstract are presented in Figure 2 and Figure 3. A network visualization map of keywords from PubMed articles (Figure 2) showed that high-frequency keywords in the field of gastrointestinal amyloidosis include terms related to organ involvement: colon (50), rectum (59), small intestine (47), stomach (47), esophagus (13), duodenum (39), jejunum (14), and liver (335); terms related to type of amyloidosis: AL amyloidosis (7), AA amyloidosis (10), and familial amyloidosis (18); terms related to diagnostics: biopsy (168), Congo Red (35), and staining and labeling (25); terms related to the clinical features of the disease: intestinal mucosa (64), gastrointestinal hemorrhage (40), diarrhea (35), gastrointestinal motility (11) abdominal pain (10), weight loss (10), esophageal achalasia (6), and constipation (5); and terms related to imagistic and functional tests: colonoscopy (19), endoscopy (17), ultrasonography (16), gastroscopy (10), hematoma (9), and atrophy (6). A network visualization map of terms from Scopus articles (Figure 3) showed that the topics in the researched field were related to the diagnosis and manifestation of amyloidosis.

Researchers’ network patterns, based on a network visualization map of co-authorship from PubMed documents, revealed collaboration between researchers on topics related to amyloidosis. One of the collective endeavors retrieved was represented by Ando Y (25 articles), Ikeda S (16), Yazaki M (14), Ueda M (10), Obayashi K (9), Fujishima M (9), Tada S (8 articles), Iida M (8) and colleagues from Japan, from institutions such as Kumamoto, Shinshu, Osaka, and Nagoya Universities. The main research topics that they analyzed were the effects of transthyretin amyloid fibrils and prefibrillar aggregates on different cellular lines, focusing on familial amyloidotic polyneuropathy and cultured cell lines [53,54]. Other collaborative works identified in this network analysis of the literature is represented by Westermark P (21 articles), Westermark GT (10 articles), and Husby G (6 articles), alongside colleagues from Sweden (from Linköping, Uppsala Universities, Ludwig Institute for Cancer Research, Karolinska Institute) and their collaborators from the USA (Argonne National Laboratory, Massachusetts Institute of Technology), Norway (Oslo University), and Japan (Kumamoto University). This collaborative group of researchers focused on topics like the molecular aspects of different classes of amyloid fibril proteins, the distribution of transthyretin-containing cells in islets of Langerhans in type 2 diabetic and nondiabetic individuals, liver transplant in amyloidosis cases or the analysis of amyloid deposits at systemic tissue sites [54,55,56,59].

Another group of authors revealed in network bibliometric analysis is represented by Skinner M (11 articles), Cohen AS (9 articles), Benson MD (6 articles), Cornwell (10 articles), Gertz (10 articles), Kyle RA (5 articles), and colleagues from the USA (from Boston University, Mayo Clinic Rochester, Hanover). The main amyloidosis topics studied by these researchers were the identification of amyloid fibrils of kappa origin after denaturation, presumed hepatic amyloidosis in patients with systemic amyloidosis, the description of islet amyloidosis and β-cell destruction in type 2 diabetes, and amyloid behavior in the senescence process [60,61,62,63] (Figure 4).

## 4. Discussion

Although current treatment options and new insights have been uncovered in the past few decades, amyloidosis remains a complex disease that requires an interdisciplinary approach [71]. The GI tract’s involvement in amyloidosis is sometimes not taken into consideration as a potential diagnosis due to its vague or atypical clinical manifestations and imagistic findings [4]. Early signs and symptoms such as unexpected weight loss, early satiety, or chronic diarrhea should initiate an endoscopic surveillance, followed by random biopsies, Congo Red stained from different segments of the GI tract, especially in the presence of a monoclonal gammopathy or signs of other organs’ involvement, such as cardiac or peripheral nervous system [72].

Compared to other GI locations, amyloid involvement in the esophagus is relatively rare and usually presents with dysphagia, as well as heartburn and occasionally even hematemesis [73]. The mechanism has yet to be fully comprehended, but some theories have been postulated throughout the years. The most appropriate ones refer to the accumulation of fibrillary deposits of amyloidogenic proteins in striated and smooth muscles or the nervous plexuses that line the GI tract [38,74]. Furthermore, Bilezikçi B et al. reported that esophageal involvement was associated with chronic inflammation and severe amyloid deposition in the blood vessel walls, which caused mucosal ischemia [40]. This suggests that amyloidogenic deposits in the esophagus may lead to functional impairment via vascular compromise rather than direct infiltration of the muscular layers. However, the study’s retrospective nature limited the authors’ ability to establish a cause–effect relationship. Esophageal amyloidosis was also reported by Shimazaki C et al. (2018) in their large cohort of systemic amyloidosis patients. However, cardiac and renal involvement were the most common findings [75].

The stomach is another possible site for the buildup of abnormally folded proteins. Usually, patients complain of abdominal pain, nausea, early satiety, gastroparesis, hematemesis, or even gastric obstruction [73,76]. Like the esophagus, stomach motility can also be affected by the accumulation of amyloid, resulting in gastroparesis, nausea, and vomiting [77]. On the other hand, gastric obstruction can also be caused by the thickening of the gastric wall secondary to fibrillary structures [78]. Park SW et al. described such a mechanism in their 2013 case report as a primary manifestation that prompted the diagnosis of amyloidosis in an 80-year-old patient. The stomach and small intestine have been mostly associated with secondary amyloidosis, as seen concerning inflammatory diseases like rheumatoid arthritis (RA) [79,80,81]. Kobayashi H et al. conducted one of the largest studies on RA-related amyloidosis. They emphasize the role of GI biopsy even without specific symptoms. Endoscopic findings ranged from gastritis to ulceration, but amyloid deposits were found in both symptomatic and asymptomatic patients [35]. Therefore, they reported that routine screening may be required to establish the GI tract’s involvement in amyloidosis in high-risk populations such as RA patients.

Amyloid involvement in the small intestine was also explored by Feurle GE et al. and Steen L. et al. while focusing on familial amyloid polyneuropathy (FAP) and its GI manifestations. Diarrhea and steatorrhea were the most common symptoms, likely due to autonomic dysfunction and amyloid infiltration in the small intestine. The study showed no correlation between the level of amyloid deposition and functional impairment such as steatorrhea and raised questions regarding the pathophysiological mechanisms involved in the symptoms. This leads to questions that future research should address to understand why patients with significant amounts of amyloid deposits remain asymptomatic while others have severe GI symptoms [46,48] (Figure 5).

The diagnosis of digestive amyloidosis could be overlooked since the clinical and imagistic findings are non-specific. One specific feature is amyloidoma, which comprises conglomerates of amyloidogenic proteins organized as tumors in different sites (which could also be found in the endoscopic exploration of the digestive tract). The pathological diagnosis of a subcutaneous fat biopsy, a skin biopsy, or tissue samples from digestive mucosa is performed using Congo Red dye. AL—light-chain amyloid; AA—insoluble serum A amyloid protein; ATTR—transthyretin-derived amyloid fibril.

(Created in BioRender)

Similar to in other locations, colonic and rectal amyloidosis can be accompanied by changes in bowel movement—diarrhea, constipation, or alternating diarrhea/constipation—and rarely by hematochezia, obstruction, or perforation [17,73,76,82,83].

Recently, Krauß L et al. (2023) published a retrospective study that drew attention to the importance of recognizing GI bleeding as a manifestation of amyloidosis and emphasized the role of biopsy for accurate diagnosis [36]. Rectal biopsy remains a very good diagnostic tool for systemic amyloidosis due to the segment’s high number of amyloid deposits. Li T et al. examined the idea of combining abdominal skin, subcutaneous fat, and rectal mucosal biopsies for diagnosing AL amyloidosis with renal involvement [69]. They found that rectal biopsies were highly sensitive at detecting amyloid deposits, supporting their use as a minimally invasive diagnostic tool. Moreover, Kuroha M et al. introduced autofluorescence imaging (AFI) endoscopy as a new diagnostic tool for detecting localized amyloidosis in the colon, which had previously gone undetected using standard endoscopy. This method showed high sensitivity, but the study was limited to only three patients. The long-term follow-up in these patients revealed no disease progression, suggesting that localized amyloidosis may have a relatively benign course in the GI tract, in contrast to systemic one [2].

In other GI locations, liver involvement in amyloidosis occurs more often than at other digestive sites [78]. Usually, it presents with jaundice, hepatomegaly, peripheral edema, and ascites, but it also can be asymptomatic [84,85]. Unfortunately, early signs are not given enough importance compared to cardiac ones; they are poor prognostic markers, and the condition can progress even to liver failure [83]. In most cases, imagistic findings show the diffuse accumulation of amyloidogenic proteins in the liver accompanied by splenomegaly and lymphadenopathy, thus showing multi-organ involvement [5,20]. A biopsy is often required to confirm the diagnosis of hepatic amyloidosis. However, some non-invasive methods have been evaluated in recent years and have proved useful for assessing liver involvement in this disease [86,87]. Brunger AF et al. showed that measuring liver stiffness could help to evaluate hepatic involvement in amyloid AL amyloidosis, thus being an alternative to avoid biopsy complications. Its prognostic values still need to be further assessed [87].

Amyloid deposits in the pancreas have been described in a few cases, and in most of these, the pancreas is not the sole organ involved. Through their research, Xin A. et al. or Jurgens CA et al. showed that amyloid deposits in pancreatic islets were associated with the loss of β-cells, thus contributing to a decline in their number and function. This was linked to the progression of type 2 diabetes [88,89]. Furthermore, Xin A et al. proved that early-onset type 2 diabetes was characterized by amyloid deposits and atrophy in the pancreas, further proving that amyloidosis may influence diabetes pathophysiology [88]. Nevertheless, endocrine involvement in amyloidosis remains an area of concern and great interest, and additional research is needed [90]. However, the scarce research describing the exocrine function of the pancreas and its involvement in GI amyloidosis makes it harder to establish a potential link between malabsorption and amyloidosis of the pancreas. The bibliometric review analysis of all documents retrieved from both databases offered an overview of the characteristics, research trends, emerging areas in the researched field, active authors in the field, patterns of collaboration between authors, and research landscapes of countries and institutions.

Some of the limitations of the studies that we included in our exhaustive research in the literature should be mentioned: (1) variability in the diagnostic techniques and criteria used across studies, which complicates the comparison of findings; (2) the presence of multiple comorbidities and treatments, which may interfere with the association between amyloidosis and GI symptoms; and (3) small sample sizes associated with retrospective designs that limit the possibility of drawing solid conclusions.

## 5. Conclusions

This systematic and bibliometric review illustrates the complexity of amyloidosis, with particular reference to its interactions with the GI tract. It emphasizes the importance of addressing GI amyloidosis, especially in clinical settings where symptoms might be overlooked. We reported GI amyloidosis’ main clinical and imaging features, focusing more on the endoscopic abnormalities associated with amyloidogenic deposits.

Amyloid deposits in the gastrointestinal (GI) tract are clinically significant due to their potential to disrupt GI function, leading to conditions like motility disorders, malabsorption, and, in severe cases, obstruction or perforation. Early and accurate diagnosis is critical, especially for high-risk individuals, such as those with chronic inflammatory conditions, as GI amyloidosis often progresses silently in its early stages. Diagnostic tools such as Congo Red staining, biopsy, and advanced imaging techniques like mass spectrometry and PET have proven valuable; however, routine endoscopy and biopsy remain central for at-risk populations.

Future research should prioritize more extensive, prospective studies to understand better the mechanisms driving organ-specific amyloid deposition and its functional impacts. There is also a clear need for developing non-invasive diagnostic methods, which would reduce the need for biopsies and enable earlier detection. Despite progress in diagnostics, treatment options for GI amyloidosis are limited. A proactive, interdisciplinary approach—including gastroenterologists, pathologists, and other specialists when needed—is essential for early recognition, targeted monitoring, and improved patient outcomes.

## Figures and Tables

**Figure 1 biomedicines-12-02630-f001:**
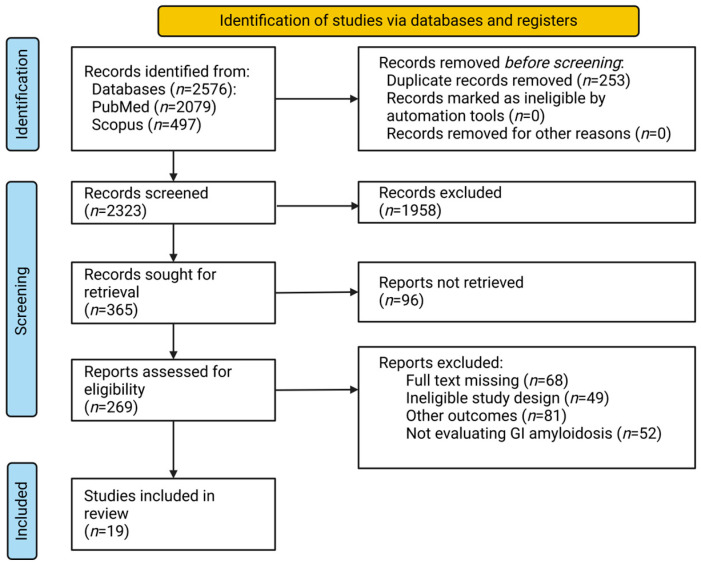
A flow diagram for this study: Preferred Reporting Items for Systematic Reviews and Meta-Analyses (PRISMA).

**Figure 2 biomedicines-12-02630-f002:**
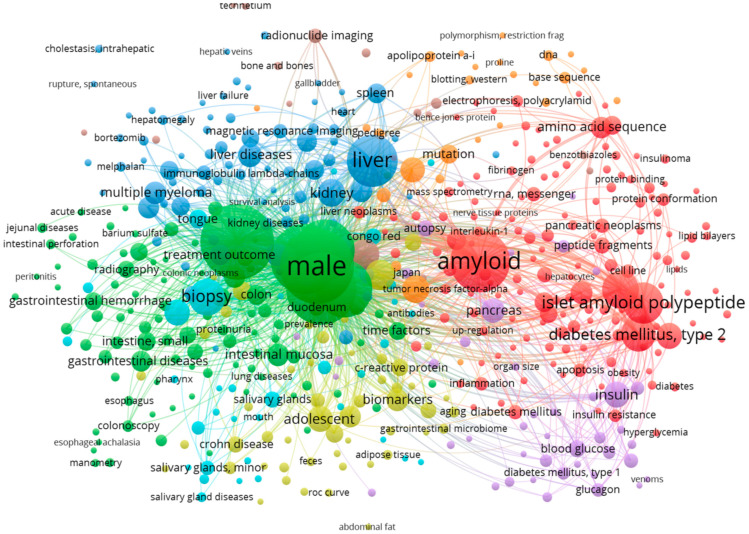
VOSviewer network visualization map of keywords from PubMed documents.

**Figure 3 biomedicines-12-02630-f003:**
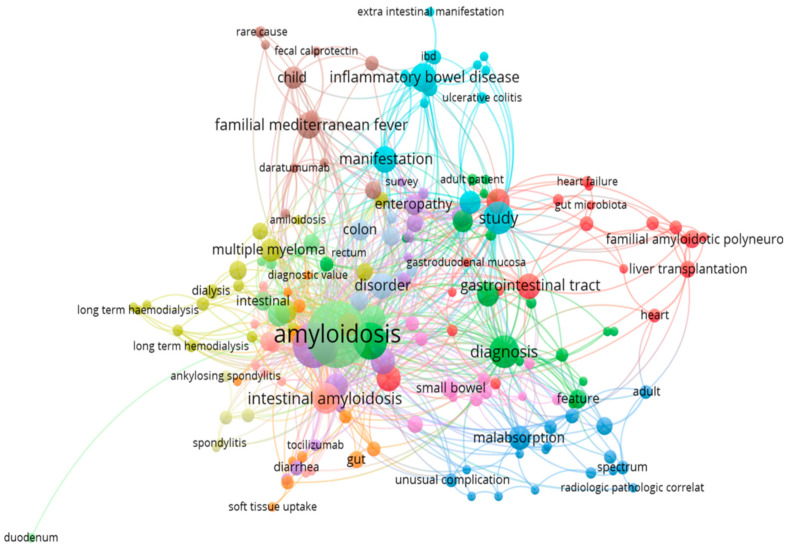
VOSviewer network visualization map of terms from the titles from Scopus documents.

**Figure 4 biomedicines-12-02630-f004:**
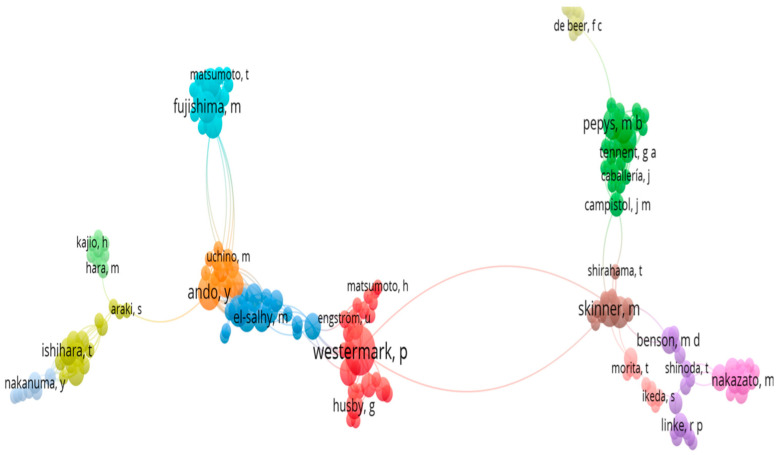
VOSviewer network visualization map of co-authorship from PubMed documents.

**Figure 5 biomedicines-12-02630-f005:**
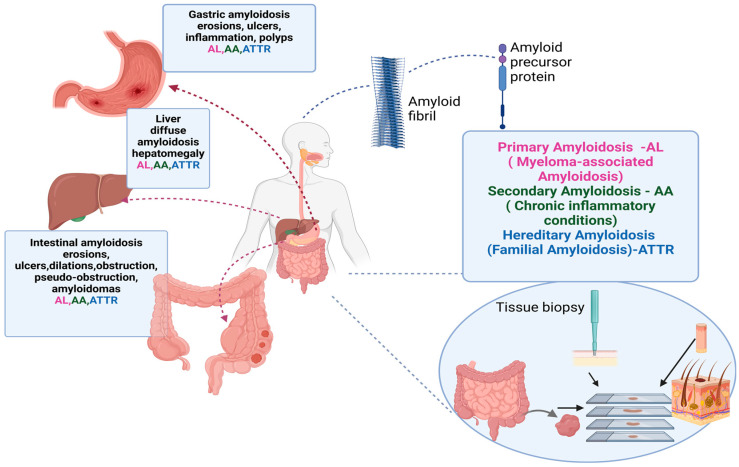
Digestive amyloidosis.

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
