# Peer review of "Digestive Amyloidosis Trends: Clinical, Pathological, and Imaging Characteristics"

_biomedicines, 2024, doi:10.3390/biomedicines12112630_

Round 1

Reviewer 1 Report

Comments and Suggestions for Authors

1.This systematic and bibliometric review suggests that all patients with myeloma, chronic inflammation, chronic infection should be screened for amyloidosis. Perhaps it is appropriate to study dialysis patients as well?

2.From this review it was possible to formulate suggested criteria for clinicians regarding the examination of patients for gastrointestinal amyloidosis.

Author Response

Dear Reviewer,

We sincerely thank you for your time and effort in reviewing our manuscript and appreciate your constructive feedback. You highlighted an essential point regarding digestive amyloidosis.

  1. This systematic and bibliometric review suggests that all patients with myeloma, chronic inflammation, chronic infection should be screened for amyloidosis. Perhaps it is appropriate to study dialysis patients as well?

Response: Amyloidosis is a complex disease that requires a multidisciplinary approach, and sometimes, the diagnosis can be tiring for both patients and clinicians. We wanted to present the data we gathered from our extensive research in the literature regarding amyloidosis clinical features and diagnostic and imagistic findings. We think that amyloidosis should also be considered in dialysis patients. However, this subject matter was not the main focus of our research.

2.From this review it was possible to formulate suggested criteria for clinicians regarding the examination of patients for gastrointestinal amyloidosis.

Response: We are glad you drew this conclusion after reading our work, as this was our primary purpose. As we all know, amyloidosis remains a challenging disease, and when it comes to the GI tract, it requires a high degree of suspicion. Through our review, we want to emphasize the main clinical and imagistic features of digestive amyloidosis to make the diagnosis and management more approachable and the background conditions that might benefit from digestive amyloidosis active screening. Still, the clinical presentation and imagistic findings remain unspecific, except for amyloidomas. 

Reviewer 2 Report

Comments and Suggestions for Authors

This review offers a comprehensive overview of gastrointestinal amyloidosis but lacks clear inclusion/exclusion criteria and supporting data, impacting the study's clarity and conclusions. The conclusion is overly verbose and should be condensed to highlight the review's key contributions and clinical implications. Additionally, there are numerous formatting errors throughout the manuscript that need to be addressed.

Title:

Abstract:

1.     The number of keywords should be reduced. (Line 35-36)

Introduction

1.     Consider streamlining the introduction of amyloidosis characteristics and its effects on different organs. (Line 39-44)

Materials and methods

1.     Clarify the 30 predefined variables mentioned in the study. (Line 89)

2.     Add the missing parenthesis around 'n=49' in the figure 1.

Result

1.     Provide a corresponding figure to illustrate the distinctive patterns of amyloid deposition in various gastrointestinal regions. (Line 167)

2.     Provide the full names of “ATTR” and “FMF” when they are first introduced in the text. (Line 182)

3.     Standardize the font style across all figure captions for uniformity. (Figure 1, 2, 3)

Discussion

1.     The descriptions regarding the liver and pancreas are not quite appropriate. Focus the topic on the gastrointestinal tract.

2.     Provide a concise explanation of why abdominal or skin fat biopsies are an alternative to repeat endoscopies and are used for diagnosing AL amyloidosis with renal involvement. (Line 218-219, 330-331)

Conclusions

1.     The conclusions section is overly verbose. Please revise it to concisely summarize the principal findings and their clinical significance.

2.     The manuscript does not explicitly explore the mechanisms of specific amyloid protein deposition. (Line 380)

3.     The limitations should be addressed in the discussion section rather than the conclusions. (Line 365-371)

4.     Ensure that the conclusions section does not reiterate content from the discussion section. (Line 372-386)

Author Response

Dear Reviewer,

We thank you for the time you committed to providing us with constructive and targeted feedback. We sincerely appreciate your comments, and we consider they will help us improve our manuscript. We carefully considered your suggestions, addressed them, and highlighted them in the manuscript.

Abstract:

The number of keywords should be reduced. (Line 35-36)

Response: We reduced the keywords according to your suggestion

Introduction

Consider streamlining the introduction of amyloidosis characteristics and its effects on different organs. (Line 39-44)

Response: We have reevaluated according to your suggestion and modified the paragraphs.

Materials and methods

  1. Clarify the 30 predefined variables mentioned in the study. (Line 89)

Response: We have clarified the content by mentioning all the variables: “consisting of the title of publication, first author, year of publication, journal of publication, type of article, number of patients, sex of the participants, age, the evolution of symptoms (months), digestive symptoms and signs, other symptoms and signs, relevant biologic markers, paraclinical diagnosis, endoscopic lesion (size, type, site), another imagistic and functional test (lesion size, site, type), tissue acquisition site and method, histological aspect, Type of disease (localized or systemic), other organs involved and which ones, type of amyloidosis. “

  1. Add the missing parenthesis around 'n=49' in the figure 1.

Response: We added the missing parenthesis

Result

  1. Provide a corresponding figure to illustrate the distinctive patterns of amyloid deposition in various gastrointestinal regions. (Line 167)

Response: Thank you for your suggestion. We added Figure 2, which illustrates the types of amyloid and amyloidosis and the involvement of the digestive system organs.

  1. Provide the full names of “ATTR” and “FMF” when they are first introduced in the text. (Line 182)

Response:  thank you for your observation; we used the full terms and then the abbreviations.

  1. Standardize the font style across all figure captions for uniformity. (Figure 1, 2, 3)

Response: We rechecked the font style across all figure captions

Discussion

  1. The descriptions regarding the liver and pancreas are not quite appropriate. Focus the topic on the gastrointestinal tract.

Response: Thank you for highlighting this point. Our focus encompasses the entire digestive system, not solely the gastrointestinal tract, and we  clarified  this in our aim and title.

  1. Provide a concise explanation of why abdominal or skin fat biopsies are an alternative to repeat endoscopies and are used for diagnosing AL amyloidosis with renal involvement. (Line 218-219, 330-331)

Response: We appreciate your kind suggestions, which helped us enrich our manuscript. Even though amyloidosis is regarded as a heterogeneous disease, the amount of research carried out in the last decades seems to shed light on its complexity. Endoscopic and imagistic techniques have helped diagnose this disease, as our literature research proves.  Moreover, whenever amyloidosis is confirmed, endoscopic surveillance should be performed to assess the progression and possible disease-associated complications. Several studies have also reported abdominal or skin fat biopsy as an alternative to repeated endoscopic procedures.

Abdominal fat pad excision biopsy (FPEB) reported good sensitivity for AL amyloidosis and higher accuracy, as the tissue sample was larger.  However, Muchtar E et al. proved that overusing organ biopsies, especially in AL amyloidosis, may fail to recognize the early stages of this disease, but further studies are needed to establish their results.  Others like Rokke HP et al. also reported good precision of abdominal fat biopsy in patients with suspected ATTR amyloidosis.  Like these methods, abdominal skin biopsy, either by excision or puncture, was also evaluated as a potential minimally invasive technique for diagnosing amyloidosis. Wu B et al. and Pinton S et al. also reported optimal sensitivity and showed promising results regarding amyloidosis diagnosis, especially for AL and ATTR types. Although large cohort studies need to be established further, these results show encouraging prospects for future research.

.

Conclusions

  1. The conclusions section is overly verbose. Please revise it to concisely summarize the principal findings and their clinical significance.

Response: Thank you for your valuable comment. We summarized the conclusion section.

  1. The manuscript does not explicitly explore the mechanisms of specific amyloid protein deposition. (Line 380)

Response:  We aim to highlight the impact of amyloidosis on the GI tract. Our extensive research revealed that there is limited information in the literature regarding the involvement of different GI segments and organs in amyloidosis. As we previously mentioned and as summarized in Table 1, our findings suggest that in some cases, there is a specific pattern in which amyloidogenic proteins tend to accumulate, leading to structures such as amyloidomas or causing distinctive granular or thickened appearances of the mucosa

  1. The limitations should be addressed in the discussion section rather than the conclusions. (Line 365-371)

Response: the limitations were addressed into the discussion section as you well suggested

  1. Ensure that the conclusions section does not reiterate content from the discussion section. (Line 372-386)

Response:  The conclusion section was modified  in a manner that does not reiterate the discussion section

Reviewer 3 Report

Comments and Suggestions for Authors

Manuscript entitled "Gastrointestinal Amyloidosis Trends: Clinical, Pathological, and Imaging Characteristics"

Major issues:

1. The authors should include the methods of detecting amyloidosis (some by HE and some by Congo red).

2. The authors should analyze the outcomes of amyloidosis in different cohorts.

3. The clinical presentation and clinical severity should also be analyzed.

4. I suggest the authors should enroll only published papers following strict criteria for the diagnosis of amyloidosis.

Author Response

Dear Reviewer,

We thank you for the time you committed to providing us with constructive and targeted feedback. We sincerely appreciate your comments, and we consider they will help us improve our manuscript. We carefully considered your suggestions and addressed them in the manuscript.

The manuscript entitled "Gastrointestinal Amyloidosis Trends: Clinical, Pathological, and Imaging Characteristics"

Major issues:

  1. The authors should include the methods of detecting amyloidosis (some by HE and some by Congo red).

Response: We added the data according to your kind recommendations.

  1. The authors should analyze the outcomes of amyloidosis in different cohorts.

Response: Most of the studies found in our search were descriptive, and the literature on this subject is scarce. Our review also emphasizes the gaps in the literature regarding digestive amyloidosis.

  1. The clinical presentation and clinical severity should also be analyzed.

Response: Thank you for your recommendation. In the Discussion section, we introduced paragraphs and analysed every digestive organ involved in GI amyloidosis, focusing on the presentation, clinical severity, and possible complications of this disease.

  1. I suggest the authors enroll only in published papers that meet strict criteria for the diagnosis of amyloidosis.

Response: Diagnosing systemic amyloidosis is not a requirement for the occurrence of digestive system amyloidosis. This review aims to highlight the gaps in the diagnosis and management of digestive amyloidosis, which may occur independently of systemic amyloidosis or other organ involvement. We emphasize that there is limited data on digestive amyloidosis, and the published literature does not provide specific characteristics or stringent criteria for its diagnosis. Most cases are identified under particular circumstances (inflammatory diseases or monoclonal gammopathy diseases) or as a diagnosis of last resort.

Round 2

Reviewer 2 Report

Comments and Suggestions for Authors

Since the last review, the authors have made significant improvements to the article, including further analysis of the data, expansion of the discussion section, and updates to the references. These enhancements have added depth and breadth to the article, increasing its academic value and clinical relevance.

However, I notice that the clarity of the VOSviewer network visualization maps (Figure 2, Figure 3 and Figure 5) in the article is insufficient, which may affect the readers' comprehension of the information presented in the figures. Please replace them.

Author Response

Dear Reviewer,

We thank you once again for your comments that helped us to improve our manuscript. We sincerely appreciate your comments:

Since the last review, the authors have made significant improvements to the article, including further analysis of the data, expansion of the discussion section, and updates to the references. These enhancements have added depth and breadth to the article, increasing its academic value and clinical relevance.

However, I notice that the clarity of the VOSviewer network visualization maps (Figure 2, Figure 3 and Figure 5) in the article is insufficient, which may affect the readers' comprehension of the information presented in the figures. Please replace them.

Response: We have replaced the figures according to your suggestion, in order to have clearer images.

Reviewer 3 Report

Comments and Suggestions for Authors

The revision is acceptable for publication

Author Response

Dear Reviewer,

We thank you for the time you committed to providing us with constructive and targeted feedback. We sincerely appreciate your comments that helped us to have an improved manuscript.